# Influence of Plant Growth-Promoting Rhizobacteria on the Formation of Apoplastic Barriers and Uptake of Water and Potassium by Wheat Plants

**DOI:** 10.3390/microorganisms11051227

**Published:** 2023-05-06

**Authors:** Zarina Akhtyamova, Elena Martynenko, Tatiana Arkhipova, Oksana Seldimirova, Ilshat Galin, Andrey Belimov, Lidiya Vysotskaya, Guzel Kudoyarova

**Affiliations:** 1Ufa Institute of Biology, Ufa Federal Research Centre, Russian Academy of Sciences, Prospekt Oktyabrya, 69, 450054 Ufa, Russia; 2Group of Culture of Beneficial Microorganisms, All-Russia Research Institute for Agricultural Microbiology, Podbelskogo sh. 3, Pushkin, 196608 Saint-Petersburg, Russia

**Keywords:** durum spring wheat, *Triticum durum* Desf., potassium, plant growth-promoting rhizobacteria (PGPR), water relations

## Abstract

The formation of apoplastic barriers is important for controlling the uptake of water and ions by plants, thereby influencing plant growth. However, the effects of plant growth-promoting bacteria on the formation of apoplastic barriers, and the relationship between these effects and the ability of bacteria to influence the content of hormones in plants, have not been sufficiently studied. The content of cytokinins, auxins and potassium, characteristics of water relations, deposition of lignin and suberin and the formation of Casparian bands in the root endodermis of durum wheat (*Triticum durum* Desf.) plants were evaluated after the introduction of the cytokinin-producing bacterium *Bacillus subtilis* IB-22 or the auxin-producing bacterium *Pseudomonas mandelii* IB-Ki14 into their rhizosphere. The experiments were carried out in laboratory conditions in pots with agrochernozem at an optimal level of illumination and watering. Both strains increased shoot biomass, leaf area and chlorophyll content in leaves. Bacteria enhanced the formation of apoplastic barriers, which were most pronounced when plants were treated with *P. mandelii* IB-Ki14. At the same time, *P. mandelii* IB-Ki14 caused no decrease in the hydraulic conductivity, while inoculation with *B. subtilis* IB-22, increased hydraulic conductivity. Cell wall lignification reduced the potassium content in the roots, but did not affect its content in the shoots of plants inoculated with *P. mandelii* IB-Ki14. Inoculation with *B. subtilis* IB-22 did not change the potassium content in the roots, but increased it in the shoots.

## 1. Introduction

The capacity of some rhizospheric bacteria to activate plant growth attracted attention of numerous researchers [1,2,3]. Stimulation of plant growth using bacteria increases plant productivity [4,5]. Therefore, it is not surprising that plant growth-promoting rhizobacteria (PGPR) are increasingly being used as inoculants to improve crop growth and yield [6], and their mechanisms of action on plant growth are being intensively investigated [6,7,8]. The stimulating effect of bacteria on plant growth can be explained through the ability of bacteria to influence the concentration of plant hormones, such as auxins, cytokinins, etc., in planta, thus affecting numerous processes in plants [9,10]. However, despite careful studies of the numerous effects of bacteria on plants, little attention was previously paid to the effects of bacteria on the formation of apoplastic barriers in plants, which play an important role in the control of the uptake of ions and water by plants, thereby affecting plant growth [11,12].

Recently, it has been shown that plant growth-promoting rhizobacteria (PGPR) (*Bacillus subtilis* IB-22 and *Pseudomonas mandelii* IB-Ki14 strains) accelerate the formation of apoplastic barriers in wheat plants subjected to salt stress, which prevents the entry of toxic sodium [13]. However, this effect may be less beneficial in the absence of stress, as it reduces the passive flow of water with solutes into the plants. Therefore, it is important to establish the relationship between the ability of bacteria to promote plant growth and their impact on the water balance.

Plants transport water and mineral elements in three ways: along the symplast (through the plasmodesmata of the cells forming the symplast), transcellularly (from cell to cell, across plasmolemma), and through apoplast [14]. Physical barriers present in exo- and endo-dermis block the apoplastic bypass flow of ions and water into the xylem, thereby reducing the uncontrolled entry of substances into the central cylinder [12,15]. At the same time, the apoplastic barriers increase the role of transport through membranes controlled through molecular carriers. As Casparian bands and suberin lamellae provide a physical barrier to apoplastic transport, ions must pass through membranes to reach the xylem [16]. Thus, barriers to the passive penetration of ions into the central cylinder along the apoplast are a necessary condition for the effective selectivity of ion uptake (and exclusion) by plants through ion channels. However, these barriers can have a negative effect on the uptake of essential elements, such as potassium.

Casparian bands form in the anticlinal cell walls between the endo- and exo-derm cells [17,18]. Isolated cell walls of the endoderm of monocots and dicots consist of suberin, lignin, carbohydrates, and structural proteins [19]. In the first stage of development of Casparian bands, endoderm cells are strongly lignified with a high content of carbohydrates and proteins, but a relatively low content of suberin [19]. Lignin is a hydrophobic compound synthesized as a result of polymerization of phenolic compounds via the phenylpropanoid pathway [20]. At a later stage of development of apoplastic barriers, suberin lamellae are also deposited on the inner surface of the radial and tangential walls of the endoderm [19,21]. The role of suberin in blocking the transport of water and solutes depends on the polyaliphatic domain, which mainly consists of oxygenated fatty acids and their derivatives [22].

Transport of water along the apoplast, bypassing cell membranes, makes a significant contribution to the total hydraulic conductivity of plant tissues. Primary cell walls have high hydraulic conductivity. However, the formation of barriers to the movement of water and ions due to their lignification, and the formation of suberin lamellae in Casparian bands, sharply reduces the hydraulic conductivity of the apoplast [12].

This problem remained unresolved if the same effect occurred in the absence of salinity, affecting the hydraulic conductivity of plants. Staining with berberine hemisulfate confirmed that bacterial treatments increased the deposition of lignin and suberin and the formation of Casparian bands in roots of barley and pea [23,24]. Calculation of hydraulic conductivity through correlating transpiration to the leaf water potential showed that it did not decrease in plants treated with bacteria. We hypothesized that the decrease in the apoplast conductance could be compensated through higher conductivity of water transport across membranes. The enhanced formation of apoplastic barriers in wheat plants treated with bacteria maintained ionic homeostasis under salinity [13]. It was of interest to find out if bacterial treatment of wheat plants, in the absence of salinity, affected the formation of apoplastic barriers and what the consequences were for the inflow of water and potassium.

Thus the aim of this work was to evaluate the effects of PGPR *Bacillus subtilis* IB-22 and *Pseudomonas mandelii* IB-Ki14 on the formation of apoplastic barriers in the inoculated durum wheat plants, and to trace the consequent effects on growth rate, hydraulic conductivity and potassium accumulation under optimal growth conditions.

## 2. Materials and Methods

### 2.1. Plant Material, Bacterial Strains and Culture Media

A gram-positive, aerobic spore-forming and cytokinin-producing strain of *Bacillus subtilis* IB-22 (GenBank MT590663) [25] and a gram-negative auxin-producing strain of *Pseudomonas mandelii* IB-Ki14 (All-Russian Collection of Microorganisms B-3250) [26] from the collection of microorganisms of the Ufa Institute of Biology of the UFIC RAS (Ufa, Russia) were used for inoculation of durum spring wheat (*Triticum durum* Desf. variety Bashkirskaya 27) plants. Bacterial inoculum was prepared via cultivating *B. subtilis* IB-22 on K1G medium containing 1% starch, 0.3% peptone, 0.3% yeast extract, 0.3% maize extract, 0.2% K_2_HPO_4_ and 0.2% (NH_4_)_2_SO_4_ and *P. mandelii* IB-Ki14-on King’s B medium (2% peptone, 1% glycerol, 0.15% K_2_HPO_4_, 0.15% MgSO_4_·7H_2_O) [27], respectively. Bacteria were cultured in Erlenmeyer flasks for 72 h at 37 °C or 48 h at 28 °C in the case of *B. subtilis* IB-22 and *P. mandelii* IB-Ki14, respectively.

### 2.2. Experimental Design

Plant growth conditions and bacterial treatments were performed as described previously [27]. To ensure drainage, a layer of gravel was placed at the bottom of 500 cm^3^ pots. After installing a glass tube for gas exchange, the pots were filled with 0.45 kg of dry soil from humus-rich clay–illuvial horizon with a medium humus content (6.3%) and supplemented with 10% sand. Wheat seeds were sterilized via soaking in a solution of 96% ethanol/3%H_2_O_2_ (1:1, *v*/*v*) for 5 min and repeatedly washed with distilled water. A total of 10 wheat seeds were placed in each vessel and inoculated with 1 mL of the bacterial suspension per seed (10^7^ CFU/mL). Plants were grown at 24 °C, irradiance of 420 μmol m^−2^ s^−1^ and in a 14 h light/10 h dark photoperiod. Plants grown in the soil without the introduced bacteria were used as a control. Soil moisture was maintained at the level of 70% of the total field capacity via watering the pots daily with distilled water. The amount of water required for irrigation was calculated via weighing the pots.

### 2.3. Visualization of Lignin and Suberin

To visualize lignin and suberin with berberine hemisulfate [28], transverse sections were manually cut from segments of the basal part of the roots with a safety razor on the sixth and eleventh days after the start of the experiments. Sections were stained with an aqueous solution of berberine hemisulfate (0.1% *w*/*v*) for 1 h and washed twice with distilled water. The sections were additionally stained for 15 min with toluidine blue (0.05% *w*/*v*) in 0.1 M phosphate buffer (pH 5.6), washed twice with distilled water, embedded in a 0.1% FeCl_3_/50% glycerol mixture and covered with a glass cover slip. Berberine fluorescence was excited with a 488 nm solid-state laser using an Olympus FluoView FV3000 confocal laser scanning confocal microscope (Olympus, Tokyo, Japan).

### 2.4. Cell Size and Leaf Area

Cell size and the area of the leaves were assessed using the program ImageJ v. 1.53t (NIH, Bethesda, MD, USA).

### 2.5. Chlorophyll Content

The relative content of chlorophyll in the epidermis of the second leaf of wheat plants was measured in vivo using DUALEX SCIENTIFIC + (FORCE-A, Orsay, France). Data were obtained on the fourteenth day after bacterial treatment.

### 2.6. Phytohormones Assay

To determine the content of phytohormones, on the sixth day after the start of the experiments, 5 plants were randomly sampled from different pots for each biological replicate (roots or shoots). Hormones were extracted with 80% ethanol (1:10) overnight. The extract separated via filtration was evaporated to an aqueous residue and divided into two parts for the determination of cytokinins and indoleacetic acid (IAA, a hormone from the class of auxins).

#### 2.6.1. Cytokinins (Content of Zeatin, Its Riboside and Nucleotide)

Cytokinins were separated from the aqueous residue via thin layer chromatography after their concentration on a C-18 cartridge (Waters Corporation, Milford, MA, USA), as previously described [29]. The content of zeatin, as well as its riboside and nucleotide in the corresponding chromatographic zones, were determined through enzyme-linked immunosorbent assay (ELISA) using antibodies against zeatin riboside [30].

#### 2.6.2. Indoleacetic Acid (IAA, a Hormone from the Class of Auxins)

IAA was partitioned from the acidified aqueous residue with diethyl ether, as previously described [26]. After methylation of samples, the IAA concentration in the extract was determined via ELISA using the corresponding specific antibodies.

### 2.7. Water Relation Measurements

#### 2.7.1. Transpiration

Transpiration was measured through comparing the weight loss of pots with 13-day-old plants, in which soil was covered for 4 h with parafilm to prevent water evaporation.

#### 2.7.2. Relative Water Content (RWC)

To determine relative water content (RWC), on the thirteenth day after the bacterial treatment, the mature first leaves of four plants were weighed and immersed in distilled water with the base; the pots were tightly closed to saturate air with water vapour and placed in the dark at room temperature. After 24 h, the turgid weight (TW) was determined after blotting, and the dry weight was determined after drying for 24 h at 80 °C. Fresh weight (FW), dry weight (DW) and TW were used to determine relative water content: RWC = (FW − DW)/(TW − DW).

#### 2.7.3. Hydraulic Conductivity 

Leaf (the middle of the first leaf) and soil water potential were measured using L-51 and PST-55 sample chambers, respectively (PSYPRO, “Wescor”, Logan, UT, USA). Hydraulic conductivity (L) was calculated, as per [27], according to the equation: L = T/[(Ys − Yl) FW], where T is transpiration, FW is the fresh weight of roots and Ys and Yl are the water potential of soil and leaves, respectively.

### 2.8. Elemental Analysis

Concentrations of sodium, potassium and phosphorus in the roots and first and second mature leaves of wheat plants were assessed using an ICPE-9000 inductively coupled plasma emission spectrometer (Shimadzu, Kyoto, Japan) [31]. For this purpose, plant samples were digested in a mixture of concentrated HNO_3_ and 38% H_2_O_2_ at 70 °C using a DigiBlock digester (LabTech, Sorisole, Italy).

### 2.9. Statistics

In figures and tables, data are presented as mean ± standard error [which were calculated using MS Excel programs]. The number of replications is provided in the figure and table legends. The significance of the differences was assessed via ANOVA, followed by Duncan’s test (*p* < 0.05) using Statistica version 10 (Statsoft, Moscow, Russia).

## 3. Results

Six days after the introduction of bacteria into the soil, there was no fluorescence of cortex cells on the transverse sections of the basal part of the roots of both the control and bacteria-treated plants (Figure 1a), which indicates a low level of lignification of their cell walls. In the region of the central cylinder, fluorescence of the cell walls was noticeable in the endodermis and xylem vessels. Root sections of plants treated with *P. mandelii* IB-Ki14 differed from plants subjected to the other treatments. For example, more intense fluorescence of the xylem vessel was noticeable, as indicated with blue- and red-coded spots, which (according to the heat map) correspond to enhanced lignification under the influence of *P. mandelii* IB-Ki14.

On the eleventh day after the start of bacterization, the pattern of fluorescence of the transverse sections of the basal part of the roots changed (Figure 1b).

Due to the fluorescence of berberine, cell boundaries in the cortex area became visible, which indicates the beginning of their lignification, although the green colour corresponds to the weak intensity of this process. In the central cylinder, the cell wall fluorescence of endodermis, xylem vessels and parenchyma increased, as evidenced via the appearance of blue and (in some sites) yellow colour. Based on the colour-coding, the fluorescence intensity increased under the influence of bacteria, especially in the case of *P. mandelii* IB-Ki14. A noticeable increase in cell size was also noted under the influence of *B. subtilis* IB-22 (Figure 2).

The introduction of bacteria into the rhizosphere of *T. durum* plants increased the weight of the shoot and the area of leaves compared to the control, which manifested their stimulating effect on plant growth (Table 1). *P. mandelii* Ki-14 did not affect the weight of plant roots, while *B. subtilis* IB-22 reduced this parameter. During the experiment, an increase in the chlorophyll content in the plants treated with either of bacteria was recorded (Table 1). 

Although the ability of these strains to produce growth-stimulating hormones was shown in a previous study [32], it was important to check how the bacteria affect the content of auxins and cytokinins in the shoots and roots of wheat plants under normal conditions. *P. mandelii* IB-Ki14, which are capable of producing IAA in vitro, increased the content of auxins in the roots by ~100% and in the shoots by ~50% compared to the control (Figure 3). Introduction of *B. subtilis* IB-22 into the rhizosphere did not influence the auxins content in either the roots or the shoots of wheat plants (Figure 3).

Introduction of the cytokinin-producing strain of *B. subtilis* IB-22 into the rhizosphere increased the total content of cytokinins in roots by ~100% and in shoots by ~50% compared to the control (Figure 4). The greatest contribution to the increase in the total content of cytokinins in the roots was made using free bases of zeatin (by 1.5 times), and in shoots using ribosides (by 2.5 times). Strain *P. mandelii* Ki-14 did not change the content of cytokinins in either roots or shoots compared to the control (Figure 4).

Accelerated growth of plants under the influence of bacteria that produce growth-stimulating hormones contributes to the formation of larger leaves that can evaporate greater quantities of water, which leads to a decrease in water content. Therefore, it was important to evaluate the effect of bacteria on the water relations of plants. Plant treatment with both studied strains led to an increase in plant transpiration calculated per whole shoot (Table 2). However, when expressed per unit of leaf area, transpiration of plants treated with *P. mandelii* Ki-14 did not differ from the control. These results indicate that when the plants were inoculated with these bacteria, the higher transpiration rate was only due to the larger leaf area from which water evaporated (Table 2). At the same time, in the case of *B. subtilis* IB-22, the increase in transpiration was recorded not only for the whole shoot, but also per leaf unit area compared to the control. This finding means that more water evaporated per leaf unit area of the inoculated plants than in the control, although transpiration increased to a lesser extent than when calculating for a whole shoot.

Inoculation with *B. subtilis* IB-22 did not affect water content in the leaves and roots, RWC or the leaf water potential, while the two last indicators were slightly decreased in the presence of *P. mandelii* IB-Ki14 (Figure 5 and Figure 6). In the plants treated with the latter strain, the decline in leaf water potential was partially compensated via decreased osmotic potential, which indicates an increase in concentration of osmotically active substances.

The calculation of hydraulic conductivity based on the results of measuring transpiration and leaf water potential through analogy with Ohm’s law showed that, in contrast to *P. mandelii* IB-Ki14, the strain *B. subtilis* IB-22 increased the ability of plant tissues to conduct water (Figure 7).

In line with our data on the acceleration and enhancement of the formation of apoplastic barriers caused by bacteria, it was of interest to analyze their effect on the accumulation of potassium, which is an essential macronutrient. 

The content of potassium in the shoots of plants inoculated with *P. mandelii* IB-Ki14 did not differ from the control; however, *B. subtilis* IB-22 increased the content of this element (Figure 8b). *P. mandelii* IB-Ki14 only reduced potassium levels in the roots (Figure 8a) at a level comparable with the control and the plants inoculated with *B. subtilis* IB-22.

## 4. Discussion

An increase in the concentration of hormones induced via PGPR that are capable of stimulating plant growth contributes to the formation of a larger leaf area, which can intensify evaporation from the surface and decrease water content in leaf tissues. Until now, little attention has been paid to this issue. Therefore, it was important to study the effects of PGPR on the growth and water relations in plants. Here the effects of *B. subtilis* IB-22 and *P. mandelii* IB-Ki14 were compared, which, as previously shown, are capable of producing cytokinins [25,32] and auxins [33], respectively. 

Present experiments confirmed that the studied bacterial strains stimulated the shoot growth of durum wheat plants and increased their leaf area. This finding was due to increased cytokinin concentration in the leaves of plants inoculated with *B. subtilis* IB-22 and increased auxins in the case of *P. mandelii* IB-Ki14. Measurement of hormone concentration in the plants confirmed that the ability of bacteria to produce hormones influenced their concentration in planta. 

The increase in the leaf area in plants inoculated with bacteria was accompanied with higher transpiration calculated at the plant level. In plants inoculated with the *B. subtilis* IB-22, increased transpiration was due not only to the greater leaf area, but apparently also to the opening of stomata, as evidenced by higher transpiration per unit leaf area. This effect may be associated with an increase in the level of cytokinins in plants, since these hormones are known to keep stomata in the open state [34]. These results are consistent with previously published findings that stated that accumulation of cytokinins in leaves increases the rate of transpiration via opening of stomata [35,36].

Despite increased transpiration, leaf hydration remained at the control level in plants treated with *B. subtilis* IB-22, and only slightly decreased when *P. mandelii* IB-Ki14 was introduced. The maintenance of tissue hydration under the influence of the *B. subtilis* IB-22 suggests that the balance between water absorption and evaporation was due to an increase in hydraulic conductivity, coupled with an increase in the transpiration. The absence of an increased hydraulic conductivity under the action of *P. mandelii* IB-Ki14 could be the reason for the above-mentioned slight decline in the water content of wheat leaf tissues. These bacteria promoted shoot growth in the wheat plants despite reduced leaf hydration, which may be due, in part, to the accumulation of osmotics in the bacteria treated plants, which helps maintain turgor. Nevertheless, the growth promoting effect of the *P. mandelii* IB-Ki14 was smaller than in the case of plant inoculation with the *B. subtilis* IB-22. This comparison shows that the capacity of *B. subtilis* IB-22 to increase the hydraulic conductivity of the plants was a more effective strategy in terms of plant growth promotion.

Water supply of plants depends on the development of the root system. Some bacteria were shown to promote root growth [37,38,39]. However, this did not happen in the present experiments, where *P. mandelii* IB-Ki14 had no effect, and *B. subtilis* IB-22 even reduced accumulation of root biomass. The latter effect may be associated with increased cytokinin content in the plants treated with the bacillary strain, since cytokinins are known to inhibit accumulation of root biomass [29]. The size of the cortex root cells increased through the treatment, which can be explained by the known effect of cytokinins on root thickening [40,41]. This effect could lead to increased root surface; however, it was not sufficient to explain the increased hydraulic conductivity of plants treated with *B. subtilis* IB-22.

The change in the hydraulic conductivity of the roots may be associated with the activity and abundance of aquaporins. Thus, bacteria can increase expression of genes encoding aquaporins in maize plants [42]. Increased aquaporin abundance was recently found in barley plants treated with *B. subtilis* IB-22 [23]. Hormone abscisic acid (ABA) is known to increase the activity of aquaporins [43], and increased ABA concentration was found in barley plants treated with *B. subtilis* IB-22 [44]. The latter effect was attributed to the capacity of this strain to produce ABA and influence ABA metabolism in the plants. These results may explain the observed increase in hydraulic conductivity of durum wheat plants treated with *B. subtilis* IB-22.

Both bacterial strains enhanced the formation of apoplastic barriers, which was most pronounced when plants were treated with *P. mandelii* IB-Ki14. The high efficiency of *P. mandelii* IB-Ki14 in this regard can be explained by their ability to synthesize IAA and increase the level of auxins in plants. It was recently shown that auxin-mediated changes in gene transcription are required to modify suberin synthesis [45]. These results suggest that the bacteria-induced auxin accumulation contributes to the accelerated deposition of suberin in the Casparian bands. There is less information on the effect of cytokinins on the formation of secondary cell walls. However, in plants with low concentrations of cytokinins, a delay in the development of protoxylem was found, which indicates the role of these hormones in the formation of secondary cell walls [46].

As mentioned above, the formation of apoplast barriers contributes to a decrease in hydraulic conductivity. Nevertheless, we did not find a decrease in hydraulic conductivity under the influence of *P. mandelii* IB-Ki14, and hydraulic conductivity even increased in plants inoculated with *B. subtilis* IB-22. Obviously, the decrease in the hydraulic conductivity of the apoplast pathway was compensated through the alternative water flow through the membranes. It was shown that hydraulic conductivity increases with an increase in transpiration [14,47]. These literature data are consistent with our results, showing that the treatment with *B. subtilis* IB-22 increases both the rate of transpiration per unit lea area (Table 2) and hydraulic conductivity (Figure 7). 

Measurement of chlorophyll content confirmed the beneficial effect of bacteria on these pigments involved in photosynthesis. Maintaining the potassium level is most important in the shoot, where it is necessary for the normal functioning of the photosynthetic apparatus [48]. Our results confirm that the accelerated formation of Casparian bands under the influence of bacteria, in general, did not have a negative effect on the accumulation of potassium ions in shoots. As in the case of water, the supply of potassium to plants was apparently carried out through channels. The decline in potassium concentration in the roots of durum wheat plants treated with *P. mandelii* IB-Ki14 may be associated with an increase in the formation of apoplast barriers in the roots. However, this effect did not decrease potassium concentration in the shoots, which was apparently related to the activity of the channels responsible for potassium delivery to the shoots. An increase in potassium levels under the influence of the *B. subtilis* IB-22 strain, which is capable of synthesizing cytokinins, corresponds to the information on a higher level of accumulation of this element in transgenic tomato plants with high content of cytokinins due to the expression of the *ipt* gene [49]. In the future, it remains to be seen through which channels this accumulation occurred, and whether there are other mechanisms for enhancing potassium translocation from root to shoot using bacteria. 

Thus, the acceleration and enhancement of the formation of apoplastic barriers under the influence of *P. mandelii* IB-Ki14 or *B. subtilis* IB-22 did not adversely affect plant growth under normal conditions. Blocking the apoplastic pathway in the treated plants appears to be compensated through activation of water and ion channels. Growth promoting effect of *B. subtilis* IB-22 was greater than that of *P. mandelii* IB-Ki14, which was apparently due the greater capacity of this strain to increase hydraulic conductivity and the supply of shoots of wheat plants with potassium.

In conclusion, present experiments confirmed that inoculation of *P. mandelii* IB-Ki14 or *B. subtilis* IB-22 strains into the rhizosphere accelerates and enhances the formation of apoplastic barriers in the roots of durum wheat plants in the absence of stress, while previous experiments revealed this effect under salinity [13]. Although apoplast barriers reduce the transport of water and ions through the apoplast, the bacteria-induced increase in deposition of lignin and suberin in apoplast did not adversely affect hydraulic conductivity and potassium levels in the shoots of treated plants. The decrease in water and ion transport through the apoplast of bacteria-treated plants could apparently be compensated through the changes in their membrane transport through the corresponding channels. The growth-promoting effect of *B. subtilis* IB-22 was greater than that of *P. mandelii* IB-Ki14, which could be due to an increase in potassium concentration in wheat plants treated with the strain, which is absent in the plants treated with *P. mandelii* IB-Ki14. Further experiments are needed to confirm the involvement of water and ion channels in the action of PGPR on wheat durum plants.

## Figures and Tables

**Figure 1 microorganisms-11-01227-f001:**
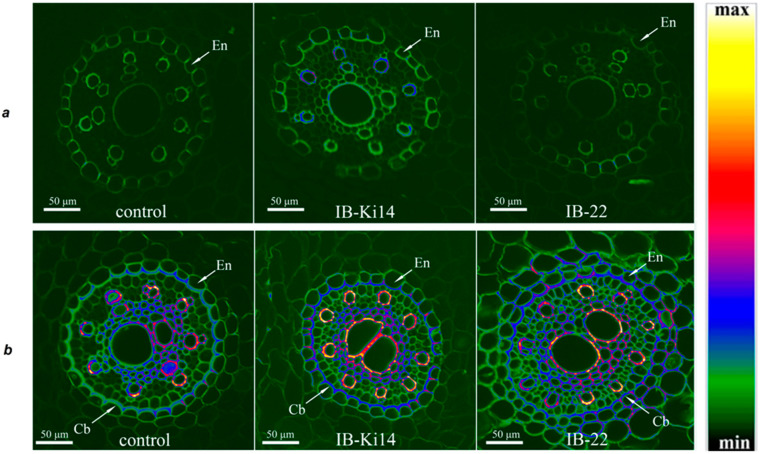
Detection of lignin and suberin via berberine fluorescence in transverse sections of basal part of *T. durum* roots 6 days (**a**) and 11 days (**b**) after bacterial treatment with *P. mandelii* IB-Ki14 or *B. subtilis* IB-22. Heatmap shows colour-coded fluorescence intensities (black/green–minimum level of lignin/suberin content; yellow/white–maximum level of lignin/suberin content). En, endoderm; Cb, Casparian bands.

**Figure 2 microorganisms-11-01227-f002:**
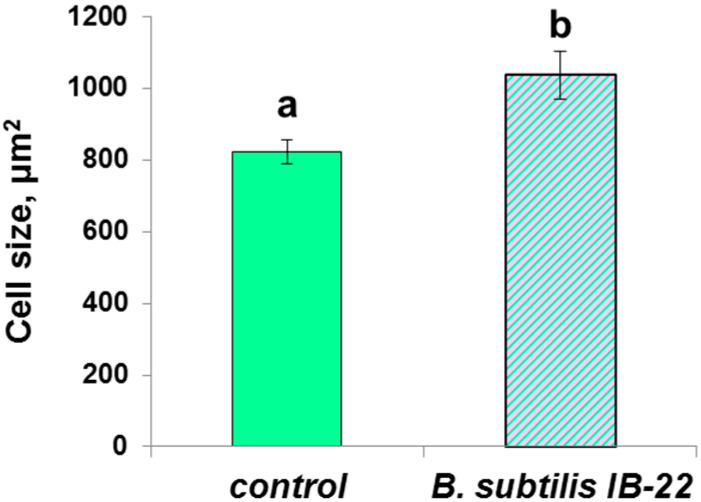
Cell size of *T. durum* plants on eleventh day after bacterial treatment with *B. subtilis* IB-22. Data are means ± SE. Significantly different means are labeled with different letters at *p* ≤ 0.05, *n* = 30 (ANOVA followed by Duncan’s test).

**Figure 3 microorganisms-11-01227-f003:**
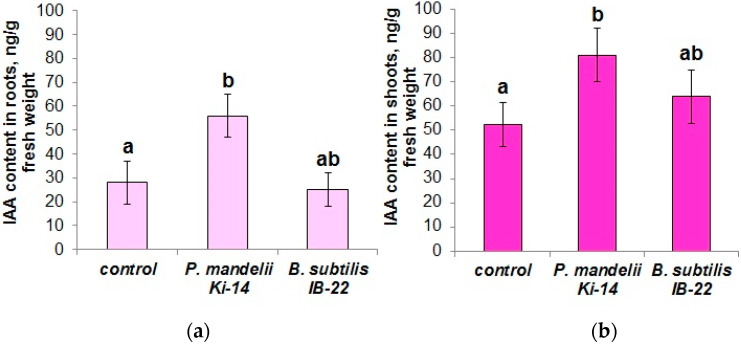
Content of IAA in roots (**a**) and shoots (**b**) of *T. durum* plants 6 days after bacterial treatment with *P. mandelii* IB-Ki14 or *B. subtilis* IB-22. Data are means ± SE. Significantly different means are labeled with different letters at *p* ≤ 0.05, *n* = 6.

**Figure 4 microorganisms-11-01227-f004:**
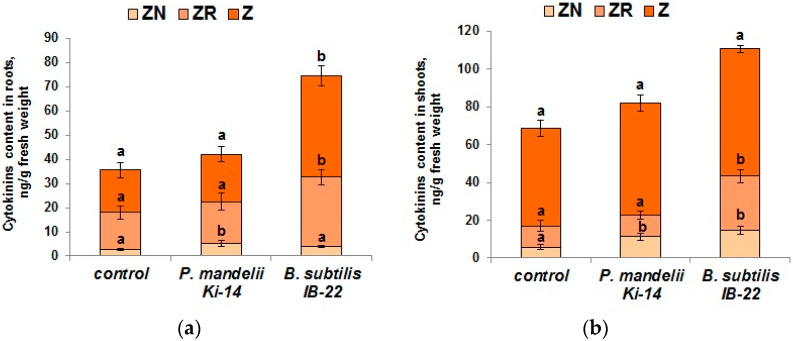
Content of cytokinins (Z—zeatin, ZN—zeatin nucleotide, ZR—zeatin riboside) in roots (**a**) and shoots (**b**) of *T. durum* plants 6 days after bacterial treatment with *P. mandelii* IB-Ki14 or *B. subtilis* IB-22. Data are means ± SE. Significantly different means are labeled with different letters at *p* ≤ 0.05, *n* = 6.

**Figure 5 microorganisms-11-01227-f005:**
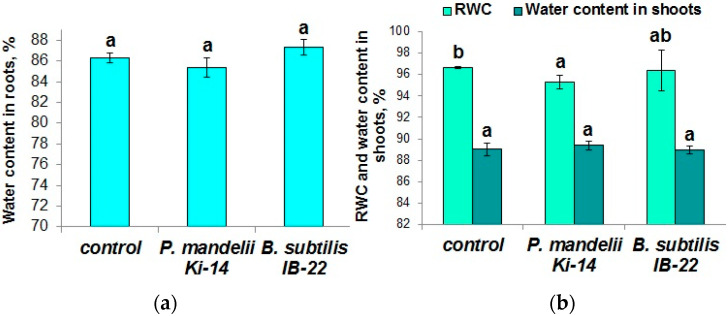
Water content in roots (**a**), RWC and water content in shoots (**b**) of *T. durum* plants 13 days after treatment with *P. mandelii* IB-Ki14 or *B. subtilis* IB-22. Data are means ± SE. Significantly different means are labeled with different letters at *p* ≤ 0.05, *n* = 5 (ANOVA followed by Duncan’s test).

**Figure 6 microorganisms-11-01227-f006:**
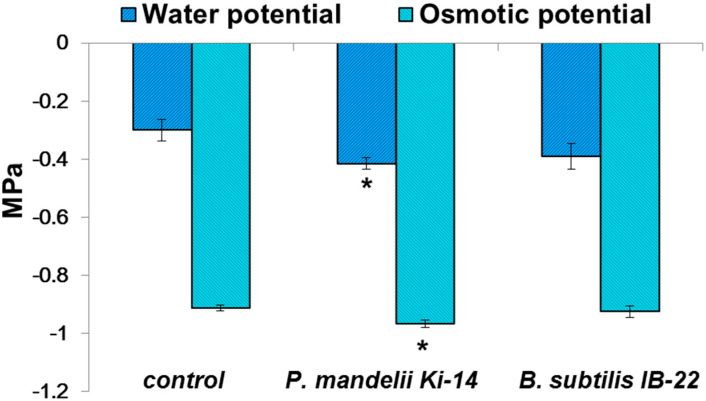
Water and osmotic potentials of *T. durum* leaves 13 days after treatment with *P. mandelii* IB-Ki14 or *B. subtilis* IB-22. * denotes the means, significantly different from the control (without bacteria) (*n* = 5, *p* ≤ 0.05, *t*-test).

**Figure 7 microorganisms-11-01227-f007:**
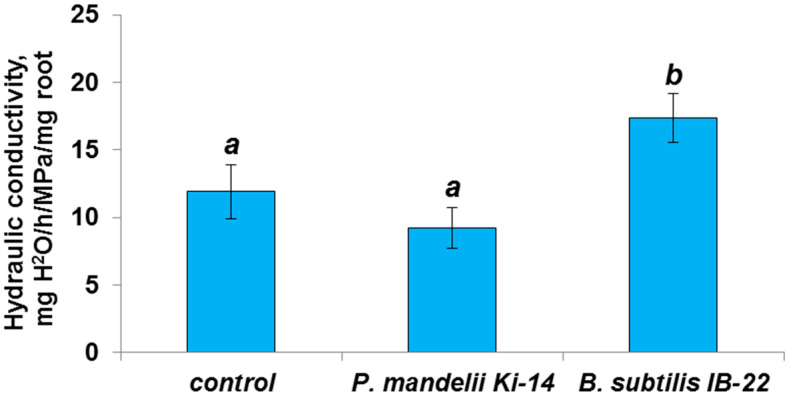
Hydraulic conductivity of *T. durum* plant roots on day 13 after seed treatment with *P. mandelii* IB-Ki14 or *B. subtilis* IB-22. Data (*n* = 6) are means ± SE. Mean values that do not differ significantly from each other are denoted by same letters (*p* ≤ 0.05, Duncan’s test).

**Figure 8 microorganisms-11-01227-f008:**
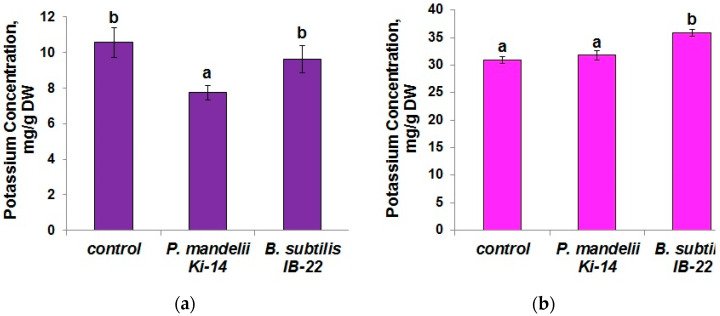
Potassium content in (**a**) roots and (**b**) shoots of *T. durum* plants 12 days after bacterial treatment with *P. mandelii* IB-Ki14 or *B. subtilis* IB-22. Data are means ± SE. Significantly different means are labeled with different letters at *p* ≤ 0.05, *n* = 6 (ANOVA followed by Duncan’s test).

**Table 1 microorganisms-11-01227-t001:** Fresh weight of roots (*n* = 10), shoots (*n* = 40), leaf area (*n* = 40) and chlorophyll content (n = 25) 14 days after introduction of bacteria into rhizosphere of *T. durum* plants.

Inoculation	Fresh Weight of Roots, mg	Fresh Weight of Shoots, mg	Leaf Area, cm^2^	Chlorophyll Content, µg/sm^2^
Control	96 ± 11 ^b^	305 ± 8 ^a^	15 ± 1 ^a^	21.4 ± 1.4 ^a^
*P. mandelii Ki-14*	85 ± 8 ^ab^	357 ± 13 ^b^	19 ± 2 ^b^	25.5 ± 1.3 ^b^
*B. subtilis IB-22*	77 ± 6 ^a^	395 ± 10 ^c^	21 ± 1 ^c^	28.2 ± 0.76 ^c^

Significantly different means are labeled with different letters at *p* ≤ 0.05 (ANOVA followed by Duncan’s test).

**Table 2 microorganisms-11-01227-t002:** Transpiration of *T. durum* plants 13 days after treatment with *P. mandelii* IB-Ki14 or *B. subtilis* IB-22.

Inoculation	Transpiration	
	mg/plant/hour	mg/cm^2^/hour
Control	175 ± 6 ^a^	11.25 ± 1.5 ^a^
*P. mandelii Ki-14*	213 ± 3 ^b^	10.94 ± 1.4 ^a^
*B. subtilis IB-22*	310 ± 12 ^c^	14.56 ± 0.8 ^b^

Significantly different means are labeled with different letters at *p* ≤ 0.05, *n* = 5 (ANOVA followed by Duncan’s test).

## Data Availability

Not applicable.

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
