# Peer review of "Influence of Plant Growth-Promoting Rhizobacteria on the Formation of Apoplastic Barriers and Uptake of Water and Potassium by Wheat Plants"

_microorganisms, 2023, doi:10.3390/microorganisms11051227_

Round 1
Reviewer 1 Report
This research topic seems exciting and appropriate for publication in Microorganisms (ISSN 2076-2607). The manuscript, titled "Influence of PGPR on the formation of apoplastic barriers and uptake of water and potassium by wheat plants" (microorganisms-2353536), explores the content of cytokinins, auxins and potassium, water relations characteristics as well as the deposition of lignin and suberin and the formation of Casparian bands in the root endodermis of durum wheat (Triticum durum Desf.) plants.
This study has some technical and structural issues that should be addressed during the revision process.
Following are the details of the comments.
1. lines 2-3: Abbreviation in the title is not recommended.
2. line 13: It would be helpful to include lines for background and research gaps.
3. lines 30-81: Literature review is narrow, and the introduction is a bit short. In addition to adding more literature, it would be better to describe more information.
4. lines 96-106: Include literature that supports the experimental design.
6. lines 165-168: The statistical section is not adequately described. Describe the analysis in detail.
7. line 388: Conclusions should be included.
Minor editing of English language required
Author Response
We are very grateful to respected reviewer for kindly reporting that topic of our article seems exciting and appropriate for publication. We highly appreciate valuable comments of the reviewer and made our best to follow them
- lines 2-3: Abbreviation in the title is not recommended.
Response: Thanks! We substituted PGPR with plant growth-promoting rhizobacteria
- line 13: It would be helpful to include lines for background and research gaps.
Response: Thanks for this valuable comment. We introduced into the Abstract that “The formation of apoplastic barriers is important for controlling the uptake of water and ions by plants thereby influencing plant growth. However, the effects of plant growth-promoting bacteria on the formation of apoplastic barriers and the relationship between these effects and the ability of bacteria to influence the content of hormones in plants have not been sufficiently studied.”
- lines 30-81: Literature review is narrow, and the introduction is a bit short. In addition to adding more literature, it would be better to describe more information.
Response: According to this valuable comment we added to the beginning of introduction: “The capacity of some rhizospheric bacteria to activate plant growth attracted attention of numerous researchers [1-3]. Stimulation of plant growth by bacteria increases plant productivity [4,5]. Therefore, it is not surprising that plant growth-promoting rhizobacteria (PGPR) are being increasingly used as inoculants to improve crop growth and yield [6], and their mechanisms of action on plant growth are being intensively investigated [6-8]. The stimulating effect of bacteria on plant growth can be explained by the ability of bacteria to influence the concentration of plant hormones (such as auxins, cytokinins) in planta thus affecting numerous processes in plants [9,10]. However, despite careful studies of the numerous effects of bacteria on plants, little attention has been paid to the effects of bacteria on the formation of apoplastic barriers in plants, which play an important role in the control of the uptake of ions and water by plants thereby affecting plant growth [11,12]”
- lines 96-106: Include literature that supports the experimental design.
Response: In accordance with this remark we added that “Plant growth conditions and bacterial treatment were as described previously [27]”.
- lines 165-168: The statistical section is not adequately described. Describe the analysis in detail.
Response: This section was rewritten. It now sound as “In figures and tables, data are presented as mean ± standard error [which were calculated using MS Excel programs]. Number of replications is provided in the figure and table legends. The significance of differences was assessed by ANOVA followed by Duncan’s test (p < 0.05) using Statistica version 10 (Statsoft, Moscow, Russia)”
- line 388: Conclusions should be included.
Response: In accordanсe with this remark we added conclusion to the article: “Present experiments confirmed that inoculation of P. mandelii IB-Ki14 or B. subtilis IB-22 strains into the rhizosphere accelerates and enhances the formation of apoplastic barriers in the roots of durum wheat plants in the absence of stress, while previous experiments revealed this effect under salinity [13]. Although apoplast barriers reduce the transport of water and ions through the apoplast, the bacteria-induced increase in deposition of lignin and suberin in apoplast did not adversely affect hydraulic conductivity and potassium levels in the shoots of treated plants. The decrease in water and ion transport through the apoplast of bacteria-treated plants, apparently, could be compensated by the changes in their membrane transport through the corresponding channels. Growth-promoting effect of B. subtilis IB-22 was greater than that of P. mandelii IB-Ki14, which can be due to an increase in potassium concentration in wheat plants treated with the strain, which is absent in the plants treated with P. mandelii IB-Ki14. Further experiments are needed to confirm the involvement of water and ion channels in the action of PGPR on wheat durum plants”.
Reviewer 2 Report
The present article entitled” Influence of PGPR on the formation of apoplastic barriers and uptake of water and potassium by wheat plants” is based on a good theme. But the presentation and English of the paper is very poor. I am highlighted several point to improve:
- When the article mainly deals with “water and potassium” then why author started the introduction with pathogen” “Penetration of pathogenic microorganisms into plants enhances lignification in the infected zone, which is one of the protective reactions of plants”
-Fig 1 and Fig 2 can be merging in one, so that readers can easily find the differences
-Line-206- Replace the word Introduction with Inoculation
-line-208- Reframe the sentence “Root weight was not affected by P. mandelii Ki-14 and decreased by B. subtilis IB-22”
-Table-1- Replace word “without bacteria” with “Control”
-Author can add the figure 4 into table-1 and line no- 214-215 can be added in continuation with line 209
The present article entitled” Influence of PGPR on the formation of apoplastic barriers and uptake of water and potassium by wheat plants” is based on a good theme. But the presentation and English of the paper is very poor. I am highlighted several point to improve:
- When the article mainly deals with “water and potassium” then why author started the introduction with pathogen” “Penetration of pathogenic microorganisms into plants enhances lignification in the infected zone, which is one of the protective reactions of plants”
-Fig 1 and Fig 2 can be merging in one, so that readers can easily find the differences
-Line-206- Replace the word Introduction with Inoculation
-line-208- Reframe the sentence “Root weight was not affected by P. mandelii Ki-14 and decreased by B. subtilis IB-22”
-Table-1- Replace word “without bacteria” with “Control”
-Author can add the figure 4 into table-1 and line no- 214-215 can be added in continuation with line 209
Author Response
We are most grateful to the respected reviewer for critical and valuable remarks, which we carefully followed.
- The present article entitled ”Influence of PGPR on the formation of apoplastic barriers and uptake of water and potassium by wheat plants” is based on a good theme. But the presentation and English of the paper is very poor.
Response: English of the text was corrected by our colleague, whose first language is English. His corrections and marked and can be tracked. To improve quality of our article we added more about the background of our article and research gaps of previous publications into the beginning of the abstract and introduction and wrote Conclusion.
- When the article mainly deals with “water and potassium” then why author started the introduction with pathogen” “Penetration of pathogenic microorganisms into plants enhances lignification in the infected zone, which is one of the protective reactions of plants”
Response: We are sorry for writing about this effects related to plant resistance to biotic stress, while our article deals with the plants growing in the absence of any stress. We just wanted to emphasize that previously published reports did not consider effects of bacteria on formation of apoplastic barriers in regard to the transport of water and ions, but considered only their importance for protection of plants against infection. But actually, we have already reported on bacterial effects on apoplastic barriers in regard to ions transport under salinity (the corresponding article of Martynenko et al., 2022 cited in the present article). So we just deleted the sentence, which the reviewer rightly found to be out of place.
- -Fig 1 and Fig 2 can be merging in one, so that readers can easily find the differences
Response: This was done
- -Line-206- Replace the word Introduction with Inoculation
Response: This was done
- -line-208- Reframe the sentence “Root weight was not affected by P. mandelii Ki-14 and decreased by B. subtilis IB-22”
Response: The sentence was rephrased in the following way: “P. mandelii Ki-14 did not affect the weight of plant roots, while B. subtilis IB-22 reduced this parameter.”
- -Table-1- Replace word “without bacteria” with “Control”
Response: “without bacteria” was replaced with “control” in both table 1 and 2.
- -Author can add the figure 4 into table-1 and line no- 214-215 can be added in continuation with line 209
Response: This was done
Round 2
Reviewer 1 Report
The manuscript has been substantially revised by the authors, and the current draft is suitable for acceptance.
Reviewer 2 Report
Authors have made significant changes in the article as per suggestions. Article can now be accepted in the present form
Authors have made significant changes in the article as per suggestions. Article can now be accepted in the present form